# Is Silver Addition to Scaffolds Based on Polycaprolactone Blended with Calcium Phosphates Able to Inhibit *Candida albicans* and *Candida auris* Adhesion and Biofilm Formation?

**DOI:** 10.3390/ijms25052784

**Published:** 2024-02-28

**Authors:** Francesca Menotti, Sara Scutera, Eleonora Maniscalco, Bartolomeo Coppola, Alessandro Bondi, Cristina Costa, Fabio Longo, Narcisa Mandras, Claudia Pagano, Lorenza Cavallo, Giuliana Banche, Mery Malandrino, Paola Palmero, Valeria Allizond

**Affiliations:** 1Department of Public Health and Pediatrics, University of Torino, 10126 Turin, Italy; francesca.menotti@unito.it (F.M.); sara.scutera@unito.it (S.S.); eleonora.maniscalco@unito.it (E.M.); alessandro.bondi@unito.it (A.B.); cristina.costa@unito.it (C.C.); f.longo@unito.it (F.L.); narcisa.mandras@unito.it (N.M.); claudiapagano10vis@gmail.com (C.P.); lorenza.cavallo@unito.it (L.C.); valeria.allizond@unito.it (V.A.); 2Department of Applied Science and Technology, Politecnico di Torino, 10129 Turin, Italy; bartolomeo.coppola@polito.it (B.C.); paola.palmero@polito.it (P.P.); 3Department of Chemistry, NIS Interdepartmental Centre, University of Torino, 10125 Turin, Italy; mery.malandrino@unito.it

**Keywords:** poly(ε-caprolactone)-based biomaterial, calcium phosphates, silver, *Candida albicans*, *Candida auris*, anti-adhesive/antifungal properties, biofilm inhibition, Saos-2 cells’ osteogenic differentiation

## Abstract

*Candida* spp. periprosthetic joint infections are rare but difficult-to-treat events, with a slow onset, unspecific symptoms or signs, and a significant relapse risk. Treatment with antifungals meets with little success, whereas prosthesis removal improves the outcome. In fact, *Candida* spp. adhere to orthopedic devices and grow forming biofilms that contribute to the persistence of this infection and relapse, and there is insufficient evidence that the use of antifungals has additional benefits for anti-biofilm activity. To date, studies on the direct antifungal activity of silver against *Candida* spp. are still scanty. Additionally, polycaprolactone (PCL), either pure or blended with calcium phosphate, could be a good candidate for the design of 3D scaffolds as engineered bone graft substitutes. Thus, the present research aimed to assess the antifungal and anti-biofilm activity of PCL-based constructs by the addition of antimicrobials, for instance, silver, against *C. albicans* and *C. auris*. The appearance of an inhibition halo around silver-functionalized PCL scaffolds for both *C. albicans* and *C. auris* was revealed, and a significant decrease in both adherent and planktonic yeasts further demonstrated the release of Ag^+^ from the 3D constructs. Due to the combined antifungal, osteoproliferative, and biodegradable properties, PCL-based 3D scaffolds enriched with silver showed good potential for bone tissue engineering and offer a promising strategy as an ideal anti-adhesive and anti-biofilm tool for the reduction in prosthetic joints of infections caused by *Candida* spp. by using antimicrobial molecule-targeted delivery.

## 1. Introduction

Worldwide, both men and women are affected by different bone disorders or traumas that represent a challenge for healthcare in an emergency setting. In these cases, orthopedic surgery is necessary but it has some drawbacks, such as infection risk, nerve damage, prolonged pain, and morbidity [1]. In natural conditions, bone tissue follows a spontaneous regeneration process based on a step-by-step deposition of new bone, thanks to the involvement of different cell types and molecular signaling [2]. Unfortunately, in an elderly population or in bone-related diseases, these physiological events are delayed and, often, the body is not able to heal the damaged bone [3]. Recently, several biomaterials have been studied for their capability to promote bone regeneration and counteract orthopedic infections. Bone substitutes displaying higher functionality, reduced costs, and antimicrobial properties are necessary. Among several polymers, polycaprolactone (PCL) is a promising biomaterial for bone regeneration in the medical field due to its numerous properties. In particular, PCL is a highly biocompatible and bioresorbable polymer that, upon its implantation, is not necessary to remove at the end of its function [4,5]. PCL is also a suitable biomaterial for producing three-dimensional (3D) constructs, called scaffolds, which are an advanced therapeutic solution for cavity-filling in bone disease or for the healing of fractured bone [6,7,8]. PCL-based devices might be produced using various methods: the salt-leaching method permits the preparation of highly porous scaffolds that can be used for long-term implants and in targeted drug release applications [5].

Moreover, the blending of this polymer with calcium phosphates, principally biphasic calcium phosphate (BCP)—a mixture of hydroxyapatite (HA) and β-tricalcium phosphate (β-TCP)—guarantees a very close similarity to the constituents of natural bone [7,9,10]. Recently, BCP has shown more suitability for orthopedic applications, compared to HA and β-TCP single phases, due to its good biocompatibility, useful mechanical characteristics, and suitable biodegradability [1]. Furthermore, its addition to a polymer matrix allows faster bone regeneration by inducing osteoblast differentiation and proliferation into the scaffold. 

In light of these advances, a growing interest in developing biomaterials for devices that are also characterized by antimicrobial properties has been reported. In fact, multidisciplinary research is addressing the development of novel delivery systems capable of releasing antimicrobials in an ideal and controlled manner, specifically via synthetic polymers loaded with molecules using different methodologies for targeted release [2]. PCL-based implants should be modified, when used as a carrier of antimicrobial substances, by adding antimicrobial agents in order to prevent microbial growth and inhibit bacterial and fungal adhesion and biofilm formation [4]. These modifications mainly comprise adding compounds with known antimicrobial characteristics, such as silver, copper, zinc oxide, and graphene, or natural substances like essential oils [5,9,11].

Prosthetic joint infections (PJIs) are serious complications after the implantation of a prosthetic device, causing extended hospital recovery times, prolonged antimicrobial therapy, and the need for revision surgery. Sometimes, they might result in the death of the patient if a resistant microorganism is the causative agent [12,13]. The most common microorganisms involved in these events are Gram-positive bacteria, such as *Staphylococcus aureus*, and coagulase-negative staphylococci, followed by Gram-negative bacteria [14,15]. Despite being infrequent, fungi can also cause PJIs, with 1–2% of cases involving *Candida* spp., and the number of occurrences has increased in recent decades [14,16]. *Candida albicans* is the most common isolated yeast found in such infections, but cases related to *C. parapsilosis*, *C. tropicalis*, and *C. glabrata* have also been reported [12,14,15,17,18]. More recently, in the orthopedic field, *C. auris* has been isolated from an intra-articular infection in a woman with a long-term ankle spacer loaded with antibacterial agents, but it has rapidly become a multidrug-resistant yeast of concern to international public health [19,20]. Because of the rarity of PJIs due to *Candida* spp., no standardized guidelines are available concerning their management, and further challenges include the provision of prompt and correct therapy and the risk of persistent infection [21]. A growing issue regarding *Candida* spp. infections is the emergence and spreading of strains resistant to the most common antifungal agents available [22]. Additionally, *C. albicans* and *C. auris* infections are also worsened by their capability to produce a stable biofilm [23]. Due to *Candida* spp. resistance to the common antifungals used in the clinical setting, international research has also focused on the use of silver against *Candida* spp. [24,25,26,27]. 

In our previous work, we established that PCL, either pure or blended with BCP, is a suitable matrix for creating 3D scaffolds able to promote osteoblast adhesion and proliferation without displaying cytotoxic behavior. Additionally, PCL could be used with silver to counteract both Gram-positive and Gram-negative bacterial growth/adhesion and biofilm formation [6,9,10,11]. However, until now, no data regarding its anti-*Candida* activities if incorporated in 3D PCL-based scaffolds re available in the scientific literature. Hence, the goal of the present research is to evaluate if a 3D scaffold based on pure PCL or blended with BCP, and with different silver concentrations, is able not only to counteract *Candida* spp. adhesion and biofilm formation but also to promote osteogenic differentiation.

## 2. Results and Discussion

### 2.1. Morphological and Chemical Characterization of the PCL-Based Constructs

#### 2.1.1. Morphological Aspects of Silver-Enriched PCL-Based Scaffolds

FESEM micrographs of the PCL (A) and 1% silver-added PCL (B) constructs are depicted in Figure 1. Samples were characterized by a significant and well-interconnected porosity, as required in biomedical scaffolds to allow effective diffusion of nutrients and colonization by eukaryotic cells. The pores show an irregular and squared morphology, the size and shape of which depend on the templating salt (NaCl, in this case) that has been used [10]. Scaffolds with or without silver addition show the same microstructural features, indicating that the antimicrobial agent plays no significant role in the structural and morphological characteristics. The microstructure of the BCP/PCL composite scaffolds are depicted in Figure 1C,D, at respectively lower and higher magnifications. Also, in this case, we can observe the negligible role of the ceramic particles in modifying the architecture of the scaffolds, which are still characterized by high porosity with a high degree of interconnectivity. Squared macropores, with a size achieving around 250 μm, can be observed. A fine and homogeneous distribution of calcium phosphate particles in the polymer matrix was also observed, as depicted in Figure 2. In this highest-magnification image, we can also observe the presence of a diffused microporosity, in the range of 1–5 μm, probably due to solvent evaporation. The presence of microporosity, in addition to the macropores, is also beneficial to scaffold integration in the living tissues, while potentially playing a role in guiding cell proliferation and differentiation and enhancing osteogenic behavior [28,29]. 

Similar microstructural features were observed for NaNO_3_-derived scaffolds where, again, the addition of silver and/or BCP particles did not affect the scaffold microstructures, which were still characterized by high porosity and interconnectivity [10].

The literature regarding 3D constructs based on pure PCL or PCL blended with BCP, performed using the salt-leaching method, is still scanty. In fact, this polymer is mainly used to prepare fibers or planar constructs. Conversely, besides the 3D structure, the scaffolds developed here are characterized by hierarchal porosity, where the micro- and macro-porosity have a synergic effect on osteointegration and osteogenesis, as required in bone tissue engineering [28,29,30]. In our previous research, we demonstrated that PCL and BCP/PCL can be added along with different antimicrobials—both silver and essential oils—without affecting their higher interconnected porosity [6,10,11]. Similarly, in research by Sadiasa and colleagues in 2014, the structure of a 3D scaffold based on poly(L-lactic acid) (PLLA) and PCL was evaluated; increased porosity and pore size were observed, particularly when PLLA was added [31]. Also, Sempertegui N. et al. (2018) demonstrated homogeneous porosity in 3D micro-porous PCL scaffolds [32]. Moreover, scaffolds based on PCL and poly (hydroxybutyrate-co-hydroxyvalerate) containing hydroxyapatite nanoparticles (HA-NPs) were fabricated, and SEM analysis revealed a uniform distribution of HA-NPs and well-structured 3D porosity [33].

#### 2.1.2. Quantification of Silver Release from the Enriched PCL-Based Constructs

The various PCL-based scaffolds were immersed in the cultural broth for different incubation time points to determine the silver’s release into the medium by inductively coupled plasma—optical emission spectroscopy (ICP-OES) analysis. As detailed in Figure 3, both the NaCl (A) and NaNO_3_ (B)-derived porous samples released silver at different loads within 12 days of immersion. In particular, boosted release was observed during the whole incubation period, with values after 12 days of incubation ranging from 9.35 mg/L to 10.4 mg/L for the samples that were pored with NaCl added at 1% and 1.2%, respectively, and ranging from 0.09 mg/L to 0.26 mg/L for the samples that were pored with NaNO_3_ added at 0.79% and 1%, respectively. These differences in silver release can be explained by the amount initially introduced into the constructs; in fact, we previously demonstrated [6] that scaffolds prepared with NaNO_3_ as the templating salt exhibited a higher silver release than those pored with NaCl; therefore, the former scaffolds were charged with a lower silver amount, making the release less sustained. The silver release data agree well with those of Qian Y. et al. (2019), who demonstrated that silver-modified/collagen-coated electrospun poly-lactic-co-glycolic acid (PLGA)/PCL scaffolds released silver over time, with a peak after 12–16 days [34]. Additionally, most of the results pertaining to the silver release were performed from electrospun fibers or membranes made of PCL; in this research, the authors revealed a tailored and progressive release of silver from these structures into the medium [35,36,37].

### 2.2. Osteointegration Properties of PCL-Based Biomaterials

Saos-2 cells can be used to evaluate the capacity to differentiate into mature osteocyte-like cells under mineralizing conditions and are an accepted human model to study in vitro the response to new biomaterials [38,39,40]. In a previous study, we revealed that silver introduced at 1.67% into PCL-based 3D scaffolds significantly affects Saos-2 cell viability [10]; thus, we reduced its concentration to ~1%. Subsequently, when the silver content was decreased, the Saos-2 cells were viable and, additionally, proliferated into the constructs [6]. For these reasons, the following experiments were performed on PCL-based specimens with the lowest silver concentrations, which were 0.79% and 1% for the 3D scaffolds pored with NaNO_3_ and NaCl, respectively. We investigated the matrix mineralization of Saos-2 in PCL scaffolds through Alizarin Red S staining after 12 and 18 days of culture, in the presence or not of osteogenic factors. As shown in Figure 4, cells on PCL scaffolds with silver content present a significant amount of matrix mineralization in the presence of osteogenic factors (ODM+), compared to cells on PCL scaffolds incubated only in DMEM (ODM−). Moreover, an increase in mineralization was observed on silver-added scaffolds, compared to the same biomaterials in the absence of silver. Although it was demonstrated that silver, and, in particular, Ag nanoparticles (AgNPs), can influence the viability of different cell types in a dose-dependent manner [41,42], it was also shown that AgNPs, when functionalized in new bone scaffold, can promote the osteoblastic differentiation of mesenchymal stromal cells (MSC), both in vitro and in vivo [43,44,45,46,47]. Here, we report that PCL scaffolds functionalized with silver, at non-toxic doses, can promote osteogenesis, indicating that these materials are promising candidates for the support of bone formation in vivo. These results agree well with those of Rezania N. et al. The authors seeded human osteoblasts (MG-63) into PCL and HA scaffolds; an MTT assay demonstrated the non-toxic behavior of the biomaterial at 7 and 14 days. SEM images showed the colonization of constructs by the cell that maintained their spherical shape, while Alizarin Red S staining revealed calcium deposition, demonstrating their differentiation [48]. Unfortunately, we could not estimate the same effect on BCP/PCL scaffolds, due to the presence of biphasic calcium phosphate in the biomaterial, which interferes with the Alizarin Red S, causing strong background positivity. Further studies are necessary to evaluate the osteogenic properties of BCP/PCL scaffolds using different assays.

### 2.3. Evaluation of the Direct Anti-C. albicans and -C. auris Effect of Silver

*C. albicans* strains were selected as fungal representative pathogens of PJIs, whereas *C. auris* ones were included since they recently emerged as a relevant concern for human health [14,16,19,20]. To assess the antifungal, anti-adhesive, and anti-biofilm properties of the PCL- and BCP/PCL-based 3D scaffolds with low silver concentrations added, different microbiological approaches were used and are presented and discussed here.

Three reference strains of *C. albicans*, namely, ATCC 10231^®^, ATCC 60193^®^, and ATCC 90028^®^, and three different isolates of *C. auris* harvested from clinical samples, specifically urine, sputum, and blood (MOL 5, MOL 8, and MOL 10, respectively), were tested for the direct anti-candida effect of silver with a microdilution assay. These *C. auris* strains were selected because they displayed a resistant profile for fluconazole (MIC ≥ 128 µg/mL).

As reported in Table 1, the vast majority of the strains displayed a susceptible pattern to silver since the MIC and MFC were <0.0012%. The silver concentrations tested by the microdilution assay were similar to those introduced into the 3D scaffolds; hence, they are reported as percentages. In fact, for PCL and BCP/PCL pored with NaCl, 1% and 1.2% of silver were added, and with NaNO_3_, 0.79% and 1% were added.

This initial screening of the direct antifungal effect of silver allowed us to select *C. albicans* ATCC 10231^®^ and *C. auris* MOL 10 for the following anti-adhesive and anti-biofilm assays.

### 2.4. Anti-Candidal Assays on PCL- and BCP/PCL-Based 3D Scaffolds

#### 2.4.1. Inhibition Halo Test Using *C. albicans* and *C. auris*

To additionally evaluate the release of silver from the 3D PCL- and BCP/PCL-based constructs and its effect on both *C. albicans* ATCC 10,231 and *C. auris* MOL 10, an inhibition halo assay was performed. In Table 2, the sizes of the inhibition diameters around the different samples are summarized. As revealed, an antifungal effect was obtained on both *C. albicans* and *C. auris* since an inhibition halo was obtained when silver was blended into the polymer, further demonstrating its release from the scaffolds. Conversely, no inhibition halo was observed for PCL-based specimens without silver, revealing that the polymer alone did not display any antifungal activity.

In Figure 5 and Figure 6, the representative pictures reporting the inhibition halo of *C. auris* observed around the samples enriched with silver are shown. As shown in Figure 5, the assay demonstrated the inhibition area around the PCL- and BCP/PCL-based biomaterials, when pored with NaCl, and with 1% and 1.2% silver added. Similarly, also shown in Figure 6, the same pattern was revealed in samples pored with NaNO_3_.

Timin A.S. et al. (2018) performed the inhibition halo assay on different types of scaffolds based on PCL and observed that PCL alone did not present an antimicrobial effect; in fact, microorganisms grew near the constructs. Conversely, the specimens with the added antimicrobial agents, ceftriaxone and SiO_2_, showed an inhibition halo [49].

#### 2.4.2. Adhesion Assays on the PCL-Based 3D Constructs against *C. albicans* and *C. auris*

To further demonstrate the effect of silver in enriched biomaterial, adhesion experiments were also performed. *C. albicans* and *C. auris* were exposed to the PCL and BCP/PCL-based 3D constructs added or not with two different silver concentrations. After 24 h, the adhered and planktonic yeasts were counted in CFU/mL. Regarding the adhered yeasts, their load on the pure PCL and BCP/PCL samples was about 10^7^ for both *C. albicans* (Figure 7A) and *C. auris* (Figure 7B). Conversely, when silver was added to the scaffolds, a significant (*p* < 0.001) decrease was obtained, with values of about 10^3^ CFU/mL for both *C. albicans* (Figure 7A) and *C. auris* (Figure 7B).

In parallel, the counts of planktonic *C. albicans* and *C. auris* exposed to the 3D constructs were also recorded. As detailed in Figure 8, after 24 h of incubation, the load of yeasts in the presence of the pure biomaterials of PCL and BCP/PCL was in the order of 10^7^ CFU/mL, but when the silver was introduced into the samples, the anti-yeast effect was revealed, with a significant decrease (*p* < 0.05) in the *C. albicans* (10^5^ CFU/mL) and *C. auris* (10^5^ CFU/mL) counts (Figure 8A,B).

To more deeply dissect the role of silver in *C. albicans* and *C. auris* characteristics when in contact with the constructs, a FESEM analysis was performed on un-sonicated samples. As reported in Figure 9, when the pure PCL-based biomaterial was not sonicated after 24 h of incubation with *C. albicans*, a well-structured biofilm was observed. Conversely, on the silver-enriched 3D constructs, only a few yeasts were seen, and they were also altered in their normal morphology. In fact, the oval-shaped yeast moved into a filamentous morphology, further indicating the silver’s effect on the *C. albicans* cells (Figure 10). A similar alteration in the morphology was also detected for *C. auris*.

The here-reported results demonstrate that pure PCL- or BCP/PCL-based scaffolds alone did not display antifungal properties. These findings are in contrast to those of Hajduga et al. (2022), who demonstrated that this polymer caused a reduction in *C. albicans* growth; however, no explanation regarding this property has been provided [5]. In fact, many other researchers are in agreement with our results [22,49], further demonstrating that PCL—either alone or coupled with BCP—did not exert antimicrobial properties. BCP biocompatibility is guaranteed by the fact that the mixtures of HA and β-TCP are close to the mineral component of natural bone [7,50]. Notably, PCL alone is not able to induce osteoblast adhesion, proliferation, and differentiation, whereas the addition of BCP and silver induced a relevant improvement in the mineralization of Saos-2 cells, due to their calcium deposition.

To our knowledge, no studies have yet evaluated the activity of a 3D scaffold based on PCL blended with BCP, and, further added with silver, against *C. albicans* and *C. auris* adhesion and biofilm formation. To this end, in the present work, silver was added to impart anti-candidal features. When PCL and BCP/PCL were functionalized with silver, a significant anti-*C. albicans* and anti-*C. auris* effect was achieved on both adhered and planktonic yeasts, additionally inhibiting biofilm formation. We also showed the direct effect of silver on the fungal cells; in fact, an alteration in their common morphology was demonstrated, and filamentous forms were detected. In full agreement with our results, the Vazquez-Munoz research group assessed the antimicrobial activity of silver nanoparticles (AgNPs) on *C. auris* planktonic and biofilm growth phases; they revealed an alteration to the usual *C. auris* shape, with elongated forms [51]. Additionally, other authors demonstrated the anti-*C. albicans* and anti-*C. auris* effects of AgNPs [25,26,52,53,54]. Mare A.D. and colleagues (2021), when analyzing AgNPs mediated by spruce bark extract (SBE), demonstrated their antifungal activity and inhibited biofilm production for *C. albicans*, *C. auris*, and *C. guilliermondii* [55]. The literature reported that silver acts in *Candida* spp. by disrupting its external layer, altering membrane permeability and some normal intracellular functions (e.g., silver ions interact with the thiol groups of enzymes and proteins, and bind DNA); here, we indirectly demonstrated these activities [24,25,26,27]. In fact, as shown by the FESEM micrographs, the yeasts’ morphology was altered by the silver’s presence in the 3D scaffolds since filamentous forms were observed instead of the physiological oval shape. Thus, we can speculate that silver interacts with the outer layers of the *Candida* spp. cell, determining the relevant modifications. Specifically, it causes membrane depolarization, cell wall disruption, augmentation in reactive oxygen species production, and enzyme inactivation [51,56].

In terms of other studies that have added antifungal compounds to PCL, several have to be mentioned. In a recent work, the authors prepared PCL-CuONPs fibers and demonstrated their antifungal activity against three different *Candida* spp. (*C. albicans*, *C. glabrata*, and *C. tropicalis*); the authors demonstrated, by SEM analysis, an alteration in the morphology of the yeasts exposed to these fibers as well [22]. The same authors revealed that free AgNPs inhibited both planktonic and biofilm-embedded *C. auris* [51]. In a research, a thin nanocomposite film made of PCL, clay mineral vermiculite, and ciclopirox olamine showed antimicrobial activity against *C. albicans*, but only after 96 h [4]. When monomethoxy poly(ethylene glycol)-poly(epsilon-caprolactone)-graft-polyethylenimine micelles were prepared and assayed for their anti-yeast effect, a decrease in planktonic cells was revealed, and they enhanced the antifungal activity against the biofilm state of *C. albicans* [57]. Finally, Rosato R. et al. (2023) evaluated *Cinnamomum cassia* essential oil, either pure or formulated in PCL nanoparticles, against ten clinical strains of *C. auris*, and showed a reduction in both candida growth and biofilm formation [19]. 

## 3. Materials and Methods

### 3.1. Fabrication and Characterization of PCL- and BCP/PCL-Based Constructs with or without Added Silver

All the samples were prepared as detailed in previous publications [6,10].

For the preparation of neat PCL samples, poly(ε-caprolactone) (PCL, Merck KGaA, Milan, Italy) pellets were solubilized in acetone (20 wt %) at 40 °C for 24 h; thereafter, granules from 2 types of inorganic salts, NaCl and NaNO_3_ (Sigma Aldrich, St. Louis, MO, USA) were sieved in the range of 125–355 μm and added as pore templating. The suspension was poured into plastic molds to produce cylindrical 3D scaffolds with adequate porosity after demolding and salt leaching, due to their immersion in deionized water for 4 days.

The fabrication of composite materials was implemented by slightly modifying the above procedure. A 70:30 weight ratio mixture of HA (Captal S BM192, Plasma Biotal Limited, Buxton, UK) and β-TCP (Captal R, Plasma Biotal Limited), which is known as biphasic calcium phosphate (BCP), was prepared and dispersed in acetone for 12 h before the addition of the polymer. A BCP/PCL 40:60 weight ratio was used.

Finally, for silver-enriched 3D scaffolds, a further modification was carried out. Silver nitrate at two different percentages with respect to PCL was dissolved in acetone, prior to BCP (in the case of composite scaffolds) and PCL addition. The silver percentages were 1 wt % and 1.2 wt % for the NaCl-derived porous constructs, and 0.79 wt % and 1 wt % for the NaNO_3_-derived constructs. Additional details on the manufacturing process of the 3D scaffolds can be found in the work of Comini S. et al. (2022) and Menotti F. et al. (2023) [6,10]. All the specimens were observed with field emission scanning electron microscopy (FESEM, Zeiss Supra 40, Jena, Germany) to evaluate the proper morphology, in terms of porosity features and BCP distribution (when added) within the polymer matrix.

### 3.2. Quantification of Silver Release from the PCL- and BCP/PCL-Based Scaffolds

The amount of silver released from the enriched PCL-based constructs over time was monitored via inductively coupled plasma–optical emission spectroscopy (ICP-OES) with a PerkinElmer Optima 7000 DV apparatus (PerkinElmer, Waltham, MA, USA). The instrument was equipped with a PEEK Mira Mist nebulizer, a cyclonic spray chamber, and an Echelle monochromator. A wavelength of 328.068 nm was used for Ag. Each concentration value was calculated using the external calibration method, with standard solutions prepared using the matrix matching method and considering the average value of 3 instrumental measurements.

### 3.3. In Vitro Mineralization of Saos-2 in PCL-Based Biomaterials with and without Silver

In previous research by our group [6], we demonstrated that Saos-2 cells were impaired in their viability when silver concentrations were at 1% and 1.2% for 3D scaffolds prepared with NaNO_3_ or NaCl, respectively, as pore agent salt. Therefore, the experiments aiming to determine the mineralization of osteoblasts were carried out using silver at 0.79% (when NaNO_3_ was used) and 1% (when NaCl was used). Saos-2 cells, cultured as previously reported [6,10,11], were seeded on PCL and BCP/PCL scaffolds at 2 × 10^4^ in 96-well plates in DMEM high glucose 10% FBS. Based on previously obtained results for cell proliferation [6], the cells were seeded on the scaffold in the presence of Ag at the lowest concentration (0.79% for scaffold pored with NaNO_3_ and 1% for scaffold pored with NaCl). After 6 days of culture, the cells were stimulated with osteogenic differentiation medium (ODM) (StemMACS™ OsteoDiff Medium, Miltenyi Biotec, Tokyo, Japan) or were left undifferentiated in DMEM 10% FBS. The medium was changed every 3 days for the entire culture period. All cell conditions were performed in triplicate. The calcium deposition from Saos-2 cells was evaluated with an Alizarin Red S assay after 12 and 18 days of culture, as previously reported [58]. Briefly, the cells were rinsed twice with PBS and then fixed with 4% PFA for 10 min at room temperature. The cells were washed with distilled water and then stained with 0.5% Alizarin Red S (Sigma Aldrich, St. Louis, MO, USA) for 30 min at room temperature. After washing in distilled water 3 times, the cells being maintained in water were observed and photographed using an optical microscope, the Leica ICC50 HD (Leica Microsystems, Tokyo, Japan). The wells were also photographed using standard photography. The cells were stored at −20 °C until photometric analysis was performed; the results were obtained after cell incubation in 10% acetic acid for 30 min. Absorbance was then measured at 405 nm with a spectrophotometer (VICTOR3TM, PerkinElmer, Waltham, MA, USA). PCL, and BCP/PCL scaffolds without cells were also stained, photographed, and analyzed as negative control conditions.

### 3.4. In Vitro Anti-C. albicans and -C. auris Assays

The in vitro antifungal experiments were performed by testing different strains of *Candida* spp., namely, *C. albicans* and *C. auris*, which are occasionally involved in orthopedic infections [14,16,19,20]. Initially, to evaluate the direct effect of silver on these pathogens, three reference strains of *C. albicans*, specifically ATCC^®^ 10231, ATCC^®^ 60193, and ATCC^®^ 90028, and three different isolates of *C. auris* harvested from clinical samples, specifically, urine, sputum, and blood (MOL 5, MOL 8, and MOL 10, respectively), were tested using a microdilution assay. In detail, the overnight cultures of each fungal strain on Sabouraud dextrose broth (SAB-B; Biokar Diagnostics, Beauvais, France) were centrifuged (4.000× *g* for 10 min) and harvested at 10^4^ colony-forming units (CFU)/mL, to achieve the same experimental conditions to the adhesion tests. Thereafter, the inoculum was placed in contact with the silver dilutions (from 2.4% to 0.0012%), and, after incubation at 37 °C for 24 h, the minimum inhibition concentration (MIC) was determined by observing the first limpid well [59,60]. Aliquots were placed and spreaded on SAB-A plates that were incubated at 37 °C for 48 h. The minimum fungicidal concentration (MFC) was defined as the lowest silver concentration that resulted in a 99.9% of reduction. When all the *Candida* spp. were tested, two strains were selected, specifically, *C. albicans* ATCC^®^ 10231 and *C. auris* MOL 10, for further microbiological assays. 

These strains were subjected to the inhibition halo (manual v 7.4; https://www.eucast.org/astoffungi/methodsinantifungalsusceptibilitytesting/susceptibility_testing_of_yeasts (accessed on 30 June 2023)), to fungal adhesion, and to biofilm observation experiments on the different PCL- and BCP/PCL-based biomaterials with added silver, according to the methodologies described in our previously research [6,10].

Briefly, for the inhibition halo test, the 0.5 McFarland (~5 × 10^6^ CFU/mL) suspensions of each yeast were uniformly spread on Sabouraud dextrose Agar (SAB-A, Becton Dickinson and Company, BD, Franklin Lakes, NJ, USA), then sterile scaffolds made of silver-added PCL- and BCP/PCL-based composites were gently placed on agar. The inhibition halo’s presence was measured (mm) around each sample after 24 h of incubation at 37 °C, demonstrating the release of silver from the added specimens and its inhibition on the *Candida* spp. growth [6,10]. Both pure PCL- and pure BCP/PCL specimens were used as control materials.

Regarding the adhesion assays, the *C. albicans* and *C. auris* strains were cultured on SAB-B overnight at 37 °C, and, after centrifugation, they were diluted to reach an inoculum of 10^4^ CFU/mL in SAB-B. Then, 7 mL of the inoculum was placed into a 6-well plate in direct contact with the different 3D scaffolds, specifically, PCL and BCP/PCL enriched or not enriched with the two silver concentrations (1% and 1.2% or 0.79% and 1% for the constructs pored with NaCl and NaNO_3_, respectively) to allow yeast adhesion into the biomaterials for 24 h at 37 °C. After incubation, the specimens were introduced into a sterile bag with 10 mL of sterile saline solution and were subjected to room-temperature sonication (Sonorex Digitec DT 31/H, BANDELIN Electronic GmbH & Co., Berlino, Germania) for 30 min in order to detach the adhered yeasts without affecting their viability. The sonication products were plated on SAB-A to quantify the adhered both *C. albicans* and *C. auris*. In parallel, the determination of the planktonic yeasts was performed by plating, after a 10-fold dilution, the broth that was in contact with the various 3D scaffolds and counting the CFU/mL [6,10]. All the adhesion experiments were performed by three different operators and repeated at least three times. Additionally, FESEM analysis was performed on the specimens before and after the overnight incubation and if they were sonicated or not.

### 3.5. Statistical Analysis

Statistical analysis was ruled out by the GraphPad Prism 9 software (San Diego, CA, USA). An unpaired Student’s *t*-test was performed to determine if there was a significant difference between the various PCL-based scaffolds that were enriched or not enriched with silver in the context of the adhesion experiments (as CFU/mL). A statistical analysis of the Alizarin Red S assay was performed using an ANOVA. The data were expressed as means and the standard error of the means. A *p*-value of <0.05 was deemed significant. 

## 4. Conclusions

The present study aimed to develop a 3D scaffold, based on PCL with calcium phosphates and added silver ions, for potential applications in bone tissue engineering. The investigated biomaterials consisted of a 3D construct made from PCL blended with BCP and enriched with silver at different concentrations. The 3D construct, characterized by highly interconnected macro-porosity within a residual micro-porosity within the struts, displayed the necessary silver release within 12 days. The constructs, thanks to the presence of both PCL as a polymer part and BCP as a calcium phosphate component, were able to slowly degrade, thus facilitating new bone synthesis. In addition, the specimens were able to counteract *C. albicans* and *C. auris* growth and adhesion, and also to inhibit their biofilm formation. Finally, 3D scaffolds in the presence of silver have osteogenic properties, improving osteo-differentiation and matrix mineralization. To the best of our knowledge, this study demonstrates for the first time the potential efficacy of BCP/PCL-based biomaterials that are enriched with silver in the fight against the adhesion and biofilm formation of *C. albicans* and *C. auris*. Hence, the herein-designed biomaterials could ensure bone tissue regeneration and implant integration, and could likewise provide antifungal properties for the composite biomaterials.

## Figures and Tables

**Figure 1 ijms-25-02784-f001:**
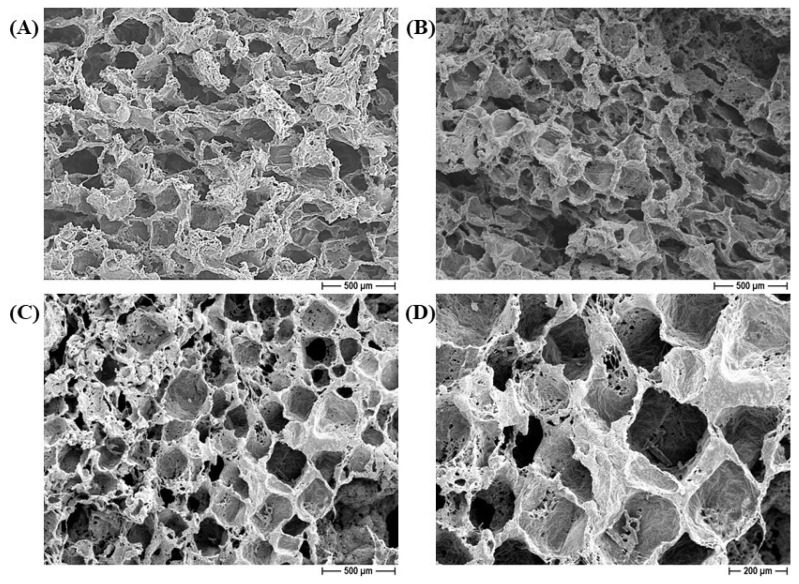
FESEM micrographs of PCL-based scaffolds: pure PCL construct, 50× (**A**), 1% silver added to a PCL-based construct, 50× (**B**), BCP/PCL construct at 50× (**C**), and 100× (**D**) magnification.

**Figure 2 ijms-25-02784-f002:**
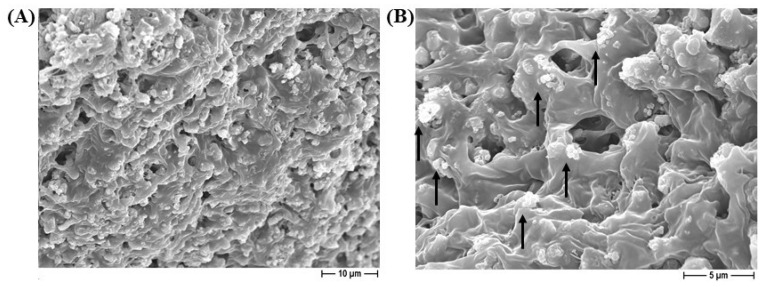
FESEM micrograph of BCP/PLC scaffold at lower (**A**) and higher (**B**) magnifications. In (**B**), the fine and homogeneous distribution of calcium phosphate particles inside the PCL polymer matrix is highlighted by the black arrows.

**Figure 3 ijms-25-02784-f003:**
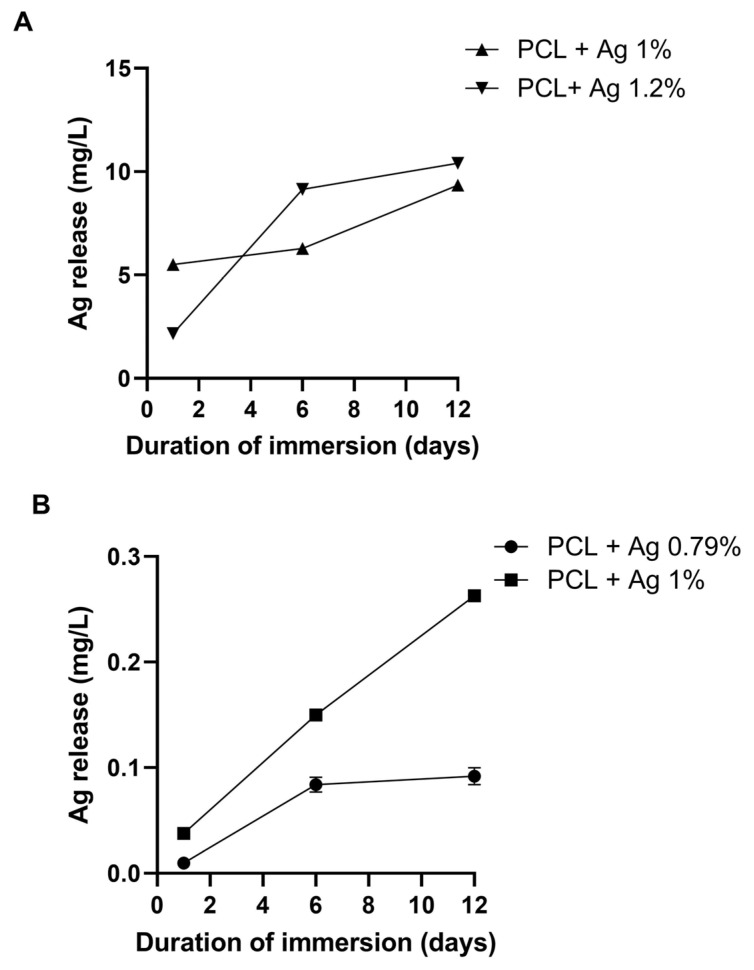
The ICP-OES analysis, revealing the silver’s release into the medium at different incubation time points from PCL-based scaffolds pored with NaCl (**A**) and NaNO_3_ (**B**) and added with the two different silver concentrations.

**Figure 4 ijms-25-02784-f004:**
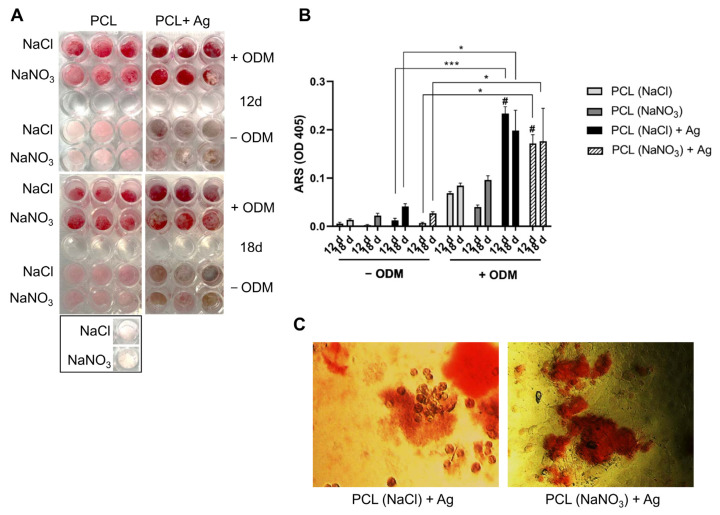
(**A**) Plate view of Alizarin Red S staining of Saos-2, cultured for 12 and 18 days in the presence or absence of ODM, on different PCL scaffolds. (**B**) Photometric quantification of Alizarin Red S staining of Saos-2 on different PCL scaffolds. * *p* < 0.05; *** *p* < 0.001; # *p* < 0.001. Saos-2 cultured on PCL + Ag scaffolds vs. cells on the respective PCL scaffolds without Ag, in the presence of ODM, are shown. Mean ± SEM of one representative experiment in triplicate. (**C**) Representative pictures of transmitted light microscopy at 20X magnification of Saos-2, cultured on PCL + Ag scaffolds and stained with Alizarin Red S to identify mineralization. ODM: osteoblast differentiation medium.

**Figure 5 ijms-25-02784-f005:**
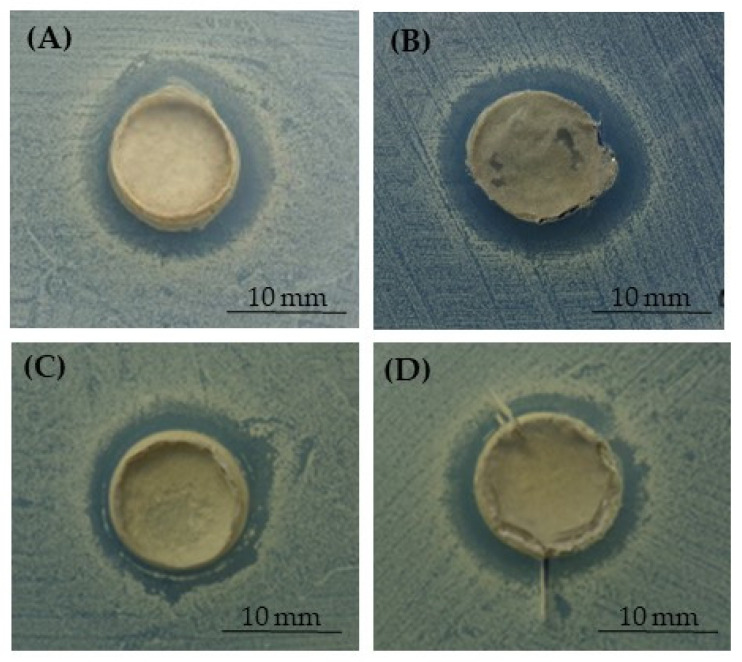
Demonstrative images of the inhibition halo assay against *C. auris* in the presence of the PCL-based samples pored with NaCl: PCL enriched with 1% (**A**) or 1.2% (**B**) of silver, and BCP/PCL enriched with 0.79% (**C**) or 1% (**D**) of silver.

**Figure 6 ijms-25-02784-f006:**
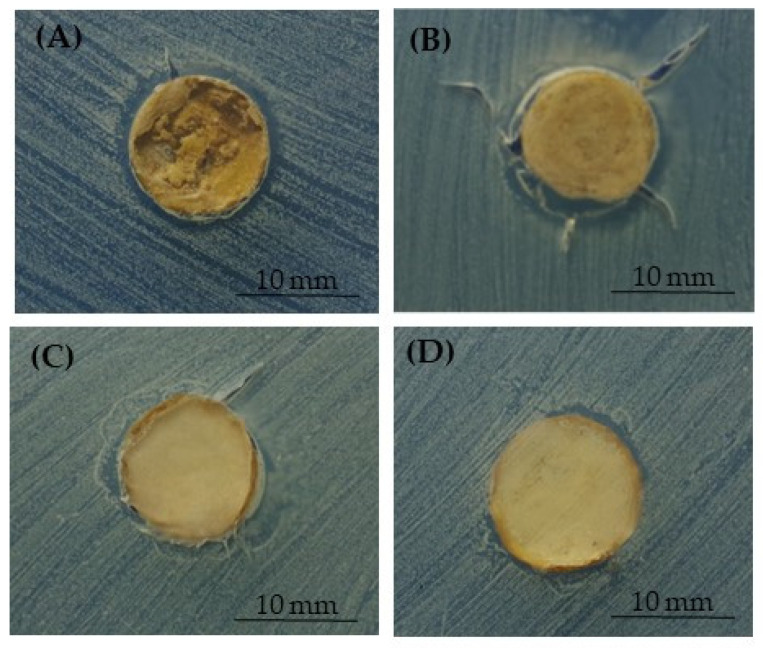
Demonstrative images of the inhibition halo assay against *C. auris* in the presence of the PCL-based samples pored with NaNO_3_: PCL enriched with 1% (**A**) or 1.2% (**B**) of silver, and BCP/PCL enriched with 0.79% (**C**) or 1% (**D**) of silver.

**Figure 7 ijms-25-02784-f007:**
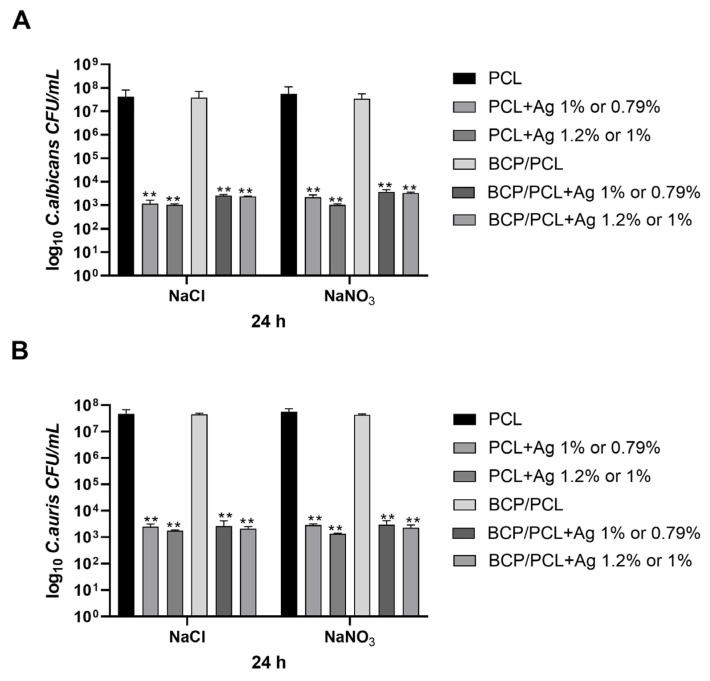
Bar charts of the number of adherent *C. albicans* (**A**) and *C. auris* (**B**) (log_10_ colony-forming unit, CFU/mL) on the PCL and BCP/PCL- based scaffolds with added lower silver concentrations, produced with either NaCl or NaNO_3_ salts, after 24 h of incubation. Results are the mean values ± standard error of the mean (SEM) of at least three independent experiments. ** *p* < 0.001 vs. PCL or BCP/PCL, via an unpaired *t*-test.

**Figure 8 ijms-25-02784-f008:**
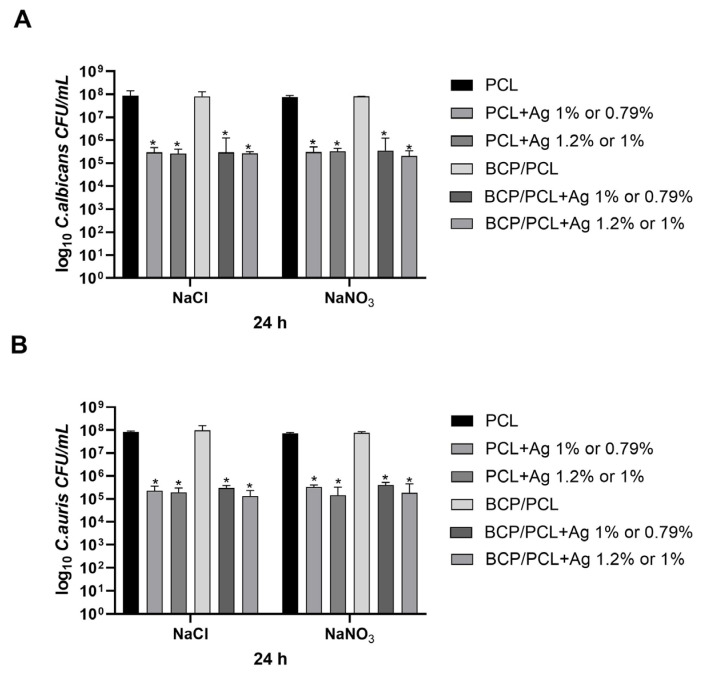
Bar charts of the number of planktonic *C. albicans* (**A**) and *C. auris* (**B**) (log_10_ colony-forming unit, CFU/mL) on the PCL and BCP/PCL-based scaffolds with added lower silver concentrations, produced with either NaCl or NaNO_3_ salts, after 24 h of incubation. Results are the mean values ± standard error of the mean (SEM) of at least three independent experiments. * *p* < 0.05 vs. PCL or BCP/PCL, via an unpaired *t*-test.

**Figure 9 ijms-25-02784-f009:**
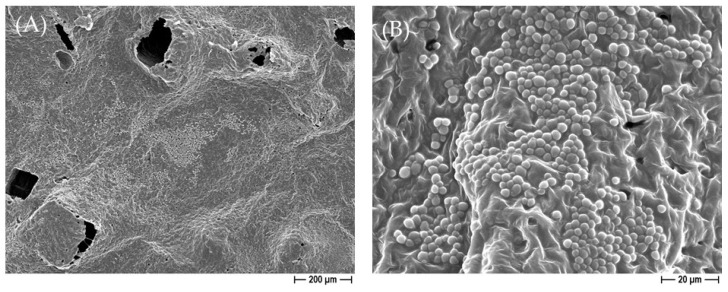
Demonstrative FESEM micrographs representing the presence of *C. albicans* biofilm on the un-sonicated pure PCL-based scaffolds, obtained by using NaCl salt as a template, at 100× (**A**) and 1000× (**B**) magnification.

**Figure 10 ijms-25-02784-f010:**
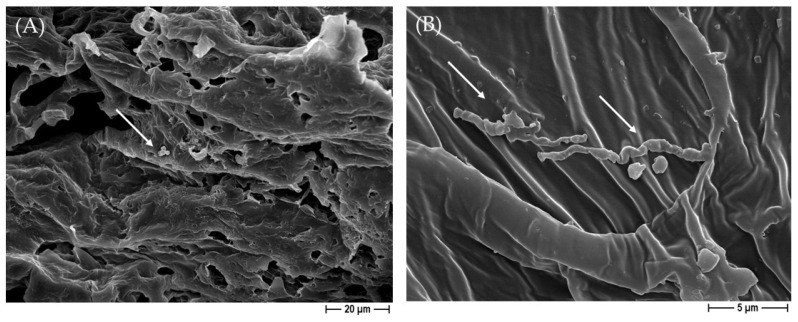
Demonstrative FESEM micrographs of the rare presence of *C. albicans* on PCL-based biomaterials, obtained by using NaCl salt as a template enriched with 1% of silver at 1000× magnification (**A**), and its morphological alteration with filamentous forms (white arrows) at 5000× magnification (**B**).

**Table 1 ijms-25-02784-t001:** Direct antifungal activity of silver on *C. albicans* and *C. auris,* expressed as MIC and MFC.

Yeast Strain	MIC % of Silver	MFC % of Silver
*C. albicans*		
ATCC 10231^®^	<0.0012%	<0.0012%
ATCC 60193^®^	<0.0012%	<0.0012%
ATCC 90028^®^	0.0024%	0.0024%
*C. auris*		
MOL 5	<0.0012%	<0.0012%
MOL 8	<0.0012%	<0.0012%
MOL 10	<0.0012%	<0.0012%

**Table 2 ijms-25-02784-t002:** Average diameters (reported as mean ± standard error of the mean) of the inhibition halo around pure PCL- or BCP/PCL-based scaffolds functionalized with low silver concentrations, pored with NaCl (A) or NaNO_3_ (B), regarding the two assayed fungal strains.

Average Diameter ± SEM (mm)		
A	*C. albicans*	*C. auris*
Scaffold Type Pored with NaCl		
PCL + Ag 1%	25.65 ± 0.18	24.12 ± 0.23
PCL + Ag 1.2%	29.01 ± 0.25	24.92 ± 0.31
BCP/PCL + Ag 1%	24.60 ± 0.21	23.98 ± 0.17
BCP/PCL + Ag 1.2%	28.02 ± 0.13	24.96 ± 0.22
B		
Scaffold Type Pored with NaNO_3_		
PCL + Ag 0.79%	24.17 ± 0.19	22.03 ± 0.37
PCL + Ag 1%	25.62 ± 0.23	24.76 ± 0.20
BCP/PCL + Ag 0.79%	24.12 ± 048	21.27 ± 0.12
BCP/PCL + Ag 1%	26.03 ± 0.31	25.06 ± 0.26

## Data Availability

The source data underlying the tables and figures are available from the authors upon request.

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
