# Peer review of "Is Silver Addition to Scaffolds Based on Polycaprolactone Blended with Calcium Phosphates Able to Inhibit Candida albicans and Candida auris Adhesion and Biofilm Formation?"

_ijms, 2024, doi:10.3390/ijms25052784_

Round 1

Reviewer 1 Report

Comments and Suggestions for Authors

After reading the manuscript, it can be published after minor revision. Authors have prepared the materials systematically. Materials characterization and discussion are also fine. I recommend the paper with minor revision....

Comments to be addressed are;

1)    How the authors confirmed the uniform dispersion of Ag with the silver enriched PCL-based scaffolds, EDS mapping would be helpful to support it.

2)    Fig 2 FESEM micrograph of BCP/PLC scaffold, showing a fine and homogeneous distribution of calcium phosphate particles inside the PCL polymer matrix… How it can be confirmed since the image is also not clear?

3)    Please command on the biocompatibility of your samples…

4)    Fig 4 , clarity of the figure may be improved

5)    “The present study aimed to develop a 3D scaffold based on PCL with calcium phosphates added with silver ions, for potential application in bone tissue engineering.” Do the authors have experimental evidence to prove the applicability of their developed material in the specified application?

Reviewer 2 Report

Comments and Suggestions for Authors

This is a relevant study that investigated the osteogenic and antimicrobial effects of a 3D scaffold model based on polycaprolactone (PCL)/biphasic calcium phosphate (PCL/BCP) associated with different concentrations of silver (Ag).

The manuscript is very clear but some considerations are relevant:

- The molecular aspects related to the physicochemical and biological properties of the materials used in the manufacture of the scaffold investigated in the study, isolated or associated (PCL-BCP-Ag), including the methodology and technology applied in preparing the scaffold, are little discussed by the Authors. A broader approach to the main factors that favor biocompatibility, regenerative capacity and antimicrobial effects would be pertinent, as well as the likely mechanisms involved in these molecular events, advantages and disadvantages in relation to other materials and technologies used.

- Another issue that should be better highlighted in "Materials and Methods" and discussed in the manuscript refers to the selection/determination of silver concentrations used in the study (concentration and toxicity).

- Furthermore, as the Authors mentioned in the text, additional experiments could complement the data obtained regarding osteogenic potential.

Round 2

Reviewer 2 Report

Comments and Suggestions for Authors

All the reviewer's concerns were sufficiently addressed by the Authors and the adjustments improved the manuscript.